# Saponins from Pea Ingredients to Innovative Sponge Cakes and Their Association with Perceived Bitterness

**DOI:** 10.3390/foods11182919

**Published:** 2022-09-19

**Authors:** Pedro Martínez Noguera, Jodie Lantoine, Even Le Roux, Suyin Yang, Ralf Jakobi, Svenja Krause, Anne Saint-Eve, Catherine Bonazzi, Barbara Rega

**Affiliations:** 1Université Paris-Saclay, INRAE, AgroParisTech, UMR SayFood, 22 Place de l’Agronomie, 91120 Palaiseau, France; 2Cargill R&D Centre Europe, Havenstraat 84, 1800 Vilvoorde, Belgium

**Keywords:** legumes, quality, flavor, taste, sensory profile, baking, reactivity, thermal degradation

## Abstract

Pea-based ingredients are increasingly being used in foods because of their nutritional, functional and environmental benefits. However, their bitter taste is not appreciated by consumers. Saponins have been reported to be bitter in whole pea flour (PF) but not in the purified ingredients obtained from it, such as pea protein isolate (PPI) and pea starch (PS). In addition, the evolution of saponins in cooked foods made from these ingredients and their relationship to bitter flavor has not been investigated. This study, therefore, explored the presence of two bitter saponins, βg and Bb, in whole pea flour (PF) and a composite flour reconstructed from the two main fractions (PS + PPI). In addition, it investigated the impact of baking on the chemical state of these compounds in a sponge cake. Finally, the sensory impact of the baking process on the perceived bitterness of cakes made with these two pea flours was also evaluated. High-Performance Liquid Chromatography–High-Resolution Mass Spectrometry (HPLC-HRMS) was used to identify and quantify pea saponins in the flours and cakes, and a descriptive sensory analysis was obtained by a trained panel to assess sensory differences in bitterness. Our results showed marked differences in saponin concentration and composition among the pea ingredients studied. Concentrations were highest in PPI (1.497 mg·g^−1^ dry matter), with 98% of saponin Bb. PS had the lowest saponin concentration (0.039 mg·g^−1^ dry matter, with 83% Bb), while 0.988 mg·g^−1^ dry matter was quantified in PF, with only 20% Bb and 80% βg. This research also highlighted the thermal degradation of saponin βg to Bb in sponge cakes during baking at 170 °C. However, at a sensory level, these chemical changes were insufficient for the impact on bitterness to be perceived in cakes made with pea flour. Moreover, baking time significantly reduced the bitter flavor in cakes made with the composite flour (PS + PPI).

## 1. Introduction

In response to the current momentum towards legume-based ingredients in the food industry, research is needed to critically assess their impacts, especially on the quality perceived by consumers in food applications. Among a diverse range of potential sources, pea (*Pisum sativum* L.) is of particular interest because of its high nutritional value, low cost, specific functional properties, and agronomic crop benefits [1,2,3]. Moreover, in view of the rapid growth of the market for gluten-free foods, alternatives to wheat flours are necessary for the increasing number of people diagnosed with celiac disease, wheat allergy, or gluten sensitivity [4]. Although several studies have reported on the potential of dry pea ingredients in traditional products [5,6,7] as well as in novel meat analogues and dairy substitutes [3,8], their unpleasant flavor still hinders widespread use, especially their green-beany aroma and bitter taste [9,10,11]. Moreover, when aiming to develop new functional plant ingredients with improved organoleptic profiles which avoid resource-intensive processing steps, a comparison of quality parameters between more and less refined ingredients is needed [12,13].

The bitter flavor of pea ingredients may be influenced by numerous chemical compounds and phenomena such as saponins [11], polyphenols [14], lipid oxidation products [15], protein hydrolysates [16], and Maillard reaction products [17]. However, saponins, non-volatile triterpenoidal glycosides, were selected for this study because of their reported sensory bitterness and thermolability [11,18,19]. In addition, along with other dry pulses, peas are one of the main sources of dietary saponins, and their health benefits are well documented [19,20]. Two main types have been described in peas: soyasaponin I or Bb (saponin Bb) and soyasaponin VI or βg (saponin βg), whose only chemical difference is a 2,3-dihydro-2,5-dihydroxy-6-methyl-4H-pyran-4-one (DDMP) attached to the aglycone C22 of saponin βg, which is substituted by a hydroxyl group in saponin Bb. It is hypothesized that βg is the native form in pea while Bb is only the product of decomposition of the former, which releases the DDMP moiety upon heating, storage, and pH changes [11,18]. Interestingly, in aqueous solutions, βg has been found to be significantly more bitter than Bb [11].

It was the thermolability of saponin βg, coupled with the loss of bittering potency, which drove this study on pea-based ingredients and sponge cakes. Although these compounds have been studied in pea as raw materials [18,21] and for canned applications [19], their characteristics in more complex foods are little understood. Sponge cakes were therefore chosen as the system to study the relationship between product quality and pea ingredient reactivity. In these studies, researchers mostly investigated odorants, while taste received less attention [7,22,23]. During the present study, two yellow pea flours, whole pea flour (PF) and a pea flour reconstituted by replacing whole pea flour with the two main purified fractions (PS + PPI, pea starch and pea protein isolate) were used for cake formulation (PFC and PSPPIC). The PS and PPI were mixed in order to maintain a constant level of the protein/starch ratio found naturally in whole pea flour, so that both flours displayed similar technological and nutritional characteristics in the cakes.

The aim of this study was threefold: first, to investigate the presence of the two saponins in both pea ingredients and sponge cakes containing pea flour and the composite pea flour; second, to elucidate the impact of baking on the chemical state of saponins quantified in PFC, and third, to explore the evolution of perceived bitterness as assessed by a trained panel at different baking times in the cakes and its link with the saponin composition of the ingredients.

## 2. Materials and Methods

### 2.1. Ingredients

Pea flour (PF), protein isolate (PPI), and starch (PS) from the same batch of yellow peas were supplied by Cargill (Vilvoorde, Belgium). PPI and PS were sub-fractions that resulted from the wet fractionation of PF [7]. The reconstructed PS + PPI flour was a 2:1 mixture (*w*/*w*) of PS and PPI. Sucrose was purchased from Tereos (Lille, France), sunflower oil from Lesieur (Asnières-sur-Seine, France), and whole pasteurized eggs came from AgroDoubs (Flagey, France).

### 2.2. Chemicals

For saponin extraction, ultrapure water obtained by a Simplicity^®^ system (Millipore, Saint Quentin, Yvelines, France), HPLC-grade ethanol (Carlo Erba, Val de Reuil, France) and Leucine-enkephalin (Sigma Aldrich, Saint Quentin Fallavier, France) were used. For HPLC analyses, deionized water, acetonitrile (Biosolve Chimie, Dieuze, France) and 99% formic acid from (Fisher Chemical, Illkirch, France) were used and were all LC/MS grade.

### 2.3. Sponge Cake Formulations

Sponge cakes were produced according to the method described by [7], which consisted of the following summarized steps: (i) eggs (45% *w*/*w*) and sucrose (25% *w*/*w*) were beaten for 10 min using a mixer equipped with a vertical whisk (KitchenAid Artisan 5KSM150, St. Joseph, MI, USA); (ii) non-sifted flours (PF or PS + PPI) (25% *w*/*w*) were gently folded into the mixture within 1.5 min; (iii) after mixing for 30 s, the sunflower oil (5% *w*/*w*) was incorporated within 15 s and the batter beaten for a further 1 min; (iv) the batter (25 g) was poured into baking molds and baked at 170 °C. Five cakes made with PF (PFC) were baked at 170 °C for 10, 15, 20, 25, and 30 min.

Using the same temperature and time points, five cakes made with PS + PPI (PSPPIC) were also produced. The cakes were labelled according to the flour type and baking time used (See Table 1).

Table 1 also details the experiments for which they were used. PFD0 was the dough analyzed as the starting point of the saponin kinetics prior to cooking. All the cakes baked were used for sensory analyses, except PDF0. The two pea flour formulations were compared after 25 min baking (PSPPIC25 and PFC25) regarding their chemical and sensory characteristics.

### 2.4. Physical Characterization of Cakes

All cakes were characterized in terms of their density, moisture content, and color. Density was determined in quadruplicate by calculating the mass-to-volume ratio. Volume was calculated using a laser-based scanner (VolScan Profiler, Stable Micro Systems, Surrey, UK). Moisture content was determined in sextuplicate by the desiccation of about 4 g, accurately weighed, of ground cake for 24 h at 105 °C in a ventilated oven (Memmert, Schwabach, Germany). Grinding was performed for 20 s at 6000 rpm using a Grindomix GM200 knife mill equipped with a stainless-steel bowl and titanium blades (Retsch GmbH, Haan, Germany). The CIE L*a*b* color parameters of the upper surface of the crust were measured at three different points using a spectro-guide sphere gloss colorimeter (BYK-Gardner, Geretsried, Germany). All physical characterization data are presented in Appendix A.

### 2.5. Saponin Analysis

#### 2.5.1. Extraction

Six hundred milligrams of either ground cake or ingredient powders (in triplicate) were extracted with 4 mL of ethanol/ultrapure water (70/30 *v*/*v*) containing 1 mg·L^−1^ Leucine-enkephalin. The latter was added as an internal standard to correct the MS detection signal over time. The suspensions were kept under constant stirring at 350 rpm at room temperature for 1 h then centrifuged at 20 °C and 3600× *g* for 10 min. The supernatants were diluted (1:10 for cake and 1:500 for ingredient powders), filtered through a 0.20 µm nylon filter, and placed in HPLC vials. These were stored at −20 °C until used for analysis.

#### 2.5.2. Identification

Pea saponins were analyzed by HPLC using a Dionex Ultimate 3000 separation module (ThermoFisher Scientific, Germering, Germany) coupled with an Orbitrap Q Exactive Focus HR-MS (ThermoFisher Scientific, Bremen, Germany). Separation was performed with an InfinityLab Poroshell 120 EC-C18 column (2.1 × 100 mm; particle diameter 2.7 µm, Agilent Technologies, Les Ulis, France). The mobile phases used were (A) deionized water containing 0.1% formic acid (*v*/*v*) and (B) acetonitrile containing 0.1% formic acid (*v*/*v*). The mobile phase flow rate was 0.5 mL·min^−1^ and the gradient program was set as follows: 20% B was held for 1 min, then B was increased from 20% to 100% within 4 min, held at 100% for 2 min and then decreased to 20% B within 0.1 min, and kept for 2.9 min for system equilibration prior to the next injection. The injection volume was 1 µL. Column temperature was set at 30 °C. MS detection was performed in negative polarity for higher sensitivity with respect to the investigated compounds. The scan type was full MS and ranged from 120.0 to 1500.0 *m/z*. The in-source CID fragmentation was 5.0 eV but no further fragmentation was applied. The monoisotopic masses of the chemical formulas of the investigated compounds were calculated: 942.5188 Da and 1068.5505 Da for saponins Bb and βg, respectively. Peak areas were calculated by extracting the chromatograms (XIC) of each pseudo molecular ion at specific mass ranges (3–5 ppm error), *m/z* 1067.5432 [M − H]^−^ and *m/z* 941.5115 [M − H]^−^ for saponins βg and saponin Bb, respectively. The peak area of Leucine-enkephalin was also obtained by integrating the peak of the specific pseudo molecular ion *m/z* 554.2611 [M − H], which then was used for peak area normalization. Identification of the two saponins was also confirmed by comparing their masses with those reported by [24] and their chemical formula were checked by looking at the specific masses and major ions.

#### 2.5.3. Quantification

External calibration curves were prepared by injecting 1 µL of the solutions prepared with the standard saponin Bb (PhytoLab, Vestenbergsgreuth, Germany) at eight different concentrations: 0.01017, 0.05085, 0.1017, 0.2034, 0.5085, 1.017, 5.085, and 10.17 mg·L^−1^ extraction solvent using the same HPLC-HRMS technique. Two sequences were run to cover the analysis of all samples prepared in triplicate and two rounds of calibration points were run at both the beginning and end of each sequence. The mean values of the normalized peak areas of saponin Bb per sequence were calculated and two linearity ranges were used to construct the regression lines. Thus, four calibration curves were obtained whose correlation coefficients were 0.9996, 0.9981, 0.9994, and 0.9979. The limit of detection (LOD) and limit of quantification (LOQ) were calculated as the signal-to-noise ratio (S/N) equal to 3 and 10, respectively, for the lowest standard concentration (0.01017 mg·L^−1^) and resulted in LOD = 0.488 µg·L^−1^ and LOQ = 1.627 µg·L^−1^. Since no standard for saponin βg is commercially available, the experimentally determined saponin βg concentrations should be considered an approximation. Results were expressed in mg of saponin per g of dry matter.

### 2.6. Descriptive Sensory Analysis

#### Sensory Evaluations

Sixteen panelists (14 female/2 male, aged between 21 and 45 years) were recruited and trained to perform a descriptive sensory analysis of the cakes. They were either students or workers at Université Paris-Saclay/UMR SayFood (AgroParisTech/INRAE). Five of the sixteen panelists had previous experience in performing sensory analyses. All samples were consumed at room temperature and were labelled with a randomized three-digit code. The whole sensory study was performed in eight sessions (one introductory session, five training sessions and two final evaluation sessions). The first four sessions were carried out in groups of eight and the final four sessions were completed in individual sensory analysis cubicles. Each training session lasted between 25 and 45 min. Reference samples were used to help the panelists identify and evaluate the sensory qualities being investigated (Table 2).

During the first training session, participants were trained to taste/smell reference samples such as caffeine (0.4–0.8 g·L^−1^), aluminum and potassium sulfate (1–2 g·L^−1^), tartaric acid (2 g·L^−1^) and sucrose (5 g·L^−1^) solutions, and a pea flour suspension (10–25 g·L^−1^). In the second session, references at different concentrations were provided to acquaint panelists with the intensity scale. This time, the reference samples were solutions of caffeine, aluminum and potassium sulfate and a suspension of pea flour in water. Based on this, over the next four sessions, panelists were trained to discriminate and rate the intensity of reference samples spread on 1 cm thick cake slices and finally rate these attributes in non-enriched cakes (PFC10, PFC15, PFC25 and PSPPIC25), in duplicate. Intensity ratings were entered manually on 10-cm unstructured line scales. During the final evaluation sessions, PFC were evaluated in the first session and PSPPIC in the second. Panelists analyzed six cakes per session because PFC25 and PSPPIC25 were duplicated in each session. Attributes were evaluated in order: first bitterness (nose clipped), then astringency and, finally, global aromatic intensity. The participants were instructed to eat a small piece of apple and rinse their mouths with water before and after eating each cake sample. Once panelists moved on to the next attribute, they were no longer able to re-evaluate the previous attribute or change their previous answers.

### 2.7. Data Analysis

Data analysis was performed using XLSTAT (version Premium 2021.2, Addinsoft, Paris, France). For saponin analysis, a one-way Analysis of Variance (ANOVA) was performed and significant differences were evaluated by Tukey’s Honest Significant Difference (HSD) post hoc test (*p* < 0.05). 

For sensory analysis, a linear complete mixed three-way ANOVA model was used, with the product, replicate, and position effects as fixed factors and panelist and product × panelist interaction effects as random factors. This initial model, Y = µ + Panelist + Product + Replicate + Product × Panelist + ɛ was simplified iteratively by removing non-significant factors until significant model parameters were obtained. Further, if a non-significant factor was included in a significant interaction, the parameter was not removed. This simplification process is detailed in Appendix C (Table A4 and Table A5). A 95% level of confidence was chosen. The post hoc test used for pairwise comparisons was Student–Newman–Keuls (SNK). 

## 3. Results

### 3.1. Saponins in Pea Ingredients and Flours

Determining the saponin composition of the raw materials was key to understanding the initial conditions and their potential to undergo chemical changes at different stages of product processing. The chemical structure of the two saponins (Figure 1a) and the measured concentrations in the pea ingredients and flours (Figure 1b) are presented in Figure 1. Numerical data can be found in Appendix B (Table A2).

In terms of the total quantified saponin concentration, PPI presented the highest concentration (1.497 mg·g^−1^ dry matter), followed by PF (0.988 mg·g^−1^ dry matter), and finally PS (0.039 mg·g^−1^ dry matter). These results indicate a greater affinity of saponins for proteins than for starch, as already reported in previous studies [25,26]. There were also significant differences in the relative abundance of these two compounds in the different ingredients studied. PPI showed a clear abundance of Bb relative to βg (98.3% of the total quantified), while conversely, in PF, the concentration of saponin βg was significantly higher than that of Bb (79.9% of total quantified). Finally, in PS, saponin Bb was observed at a larger proportion than βg (83.3% of total quantified). The abundance of βg relative to Bb in PF seemed logical in gently processed flours (dry milling only), as βg is considered to be the natural precursor of Bb [10,17]. Moreover, since PS and PPI were products of the wet fractionation of this type of PF [6], the increase in Bb and decrease in βg in these fractions may have reflected the impact of the processing method on these phytochemicals, which appeared to drive the decomposition of βg into Bb. Elevated temperatures and pH changes, both factors involved in the fractionation of PF, have been reported to potentially trigger this reaction [18]. However, matrix effects during extraction, the selective partitioning of saponins during fractionation, and the increased extractability of βg due to its higher hydrophobicity (only predicted so far) cannot be dismissed in the interpretations. Recovery studies would be useful to better understand the effects of the matrix during extraction.

### 3.2. Saponins in Pea-Based Cakes

The characterization of saponins in complex pea-based food products is an important step to better understand the impact of the food matrix and different processing conditions on these compounds. Reliable characterization should enable the assessment of the sensory impact of potential chemical changes. First, to determine whether the saponin ratios found in the flours (Figure 1a) were maintained in the cakes, the saponin composition was investigated in cakes baked under standard conditions (25 min at 170 °C) and made with both types of flour: PFC25 and PSPPIC25. The results are presented in Figure 2.

The data presented in Figure 2 confirm significantly different saponin compositions in these two cakes. PFC25 had a significant saponin βg concentration (≈22% of the total quantified), whereas the βg level in PSPPIC25 was relatively insignificant (≈1% of the total quantified). Furthermore, the total saponin concentration in PFC25 (0.236 mg saponin·g^−1^ dry matter) was significantly higher than in PSPPIC25 (0.210 mg saponin·g^−1^ dry matter), although the total amounts found in these two cakes were closer than those found in the respective ingredients (Figure 1b). In addition, the saponin ratios in the cakes differed significantly from those observed in the flours (Figure 1b). While in PF the relative abundances (% concentration of each compound to the total quantified) of Bb and βg were 79.9% and 20.1%, respectively, in PFC they changed drastically to 21.98% and 78.02%. In PS + PPI, the βg and Bb concentrations were 11.69% and 88.31% relative to the total saponin content, whereas in PSPPIC25, βg and Bb were 0.94% and 99.06%, respectively. Apart from the hypothesis that matrix effects might be partly responsible for these disparities, the cake manufacturing process seems to have strongly influenced this profile and globally altered the Bb: βg ratio. This could be due in particular to the thermolability of βg, previously demonstrated in pure ingredients [11] and here observed for the first time in a food application after applying a controlled thermal heat treatment. Finally, saponin concentrations were significantly lower in the cakes than in the flours alone by a factor of 4–5; this was expected because of the presence of other ingredients in the cake formula.

Given the higher concentration of saponin βg in the pea flour cake (PFC), this was selected to study the kinetic changes in saponin concentrations during the baking time at 170 °C. Figure 3 shows the amount of saponins obtained from the dough and cakes baked at five different baking times (Figure 3a) and the relative Bb:βg ratios (Figure 3b).

Firstly, we observed a slight increase in the total amount of saponins in line with baking time, despite the same initial concentration from the PF ingredient. These small differences may have reflected the matrix effect during the saponin extraction process, especially in the dough (PFD0), which was the wettest and an uncooked sample (Table 1), implying a different state of proteins and starch. Most importantly, we observed how the βg levels fell gradually over the baking time, while Bb increased. Plotting the Bb:βg ratio versus baking time made it possible to visualize this clear trend; it reinforced the hypothesis of the thermal degradation of βg which, upon heating, released 2,3-dihydro-2,5-dihydroxy-6-methyl-4H-pyran-4-one (DDMP) in the form of maltol and transformed into saponin Bb [18,19].

The decomposition of βg during baking, the clear differences in saponin composition between PFC25 and PSPPIC, and the reported higher bitter potency of saponin βg compared to Bb [11] led us to explore the sensory consequences of these differences regarding the bitter flavor of pea flour cakes. The numerical data used to plot Figure 2 and Figure 3 are presented in Appendix B (Table A3).

### 3.3. Sensory Evaluation of Pea-Based Cakes

Given the chemical differences observed during the first part of this study, the next question to consider was whether baking time also impacted the bitter flavor of these cakes. The PFC and PSPPIC cakes were therefore subjected to a quantitative descriptive sensory analysis. They were obtained with five different baking times at 170 °C: PFC10, PFC15, PFC20, PFC25, PFC30, PSPPIC10, PSPPIC15, PSPPIC20, PSPPIC25, and PSPPIC30. These cakes were prepared and baked following the same protocol and using the same batch of raw materials as the cakes generated for the chemical analyses (Table 1). This made it possible to obtain reproducible batches for the two studies.

#### 3.3.1. Panel Performance

Assessing the performance of panelists during the sensory evaluation of the cakes was essential to understand the homogeneity and reproducibility of the data collected. As presented in Table 1, only three attributes were evaluated by the panelists. A reliable assessment of perceived bitterness was the aim of the sensory study, while astringency and global aromatic intensity were introduced to avoid and assess possible flavor–taste interactions. As PFC was evaluated separately from PSPPIC in two different sessions, statistical analyses were also performed in accordance with the experimental design. In this way, no comparison between PFC and PSPPIC could be inferred.

Performance is detailed in Table 3 and Table 4. For PFC (Table 3), a replicate effect was not significant for any of the attributes, meaning that the panelists were repeatable in terms of rating the products. In addition, the interaction between panelists and products was not significant in all cases, indicating a positive level of agreement between panelists during the assessment. The panelist effect was significant for bitterness, astringency, and global aromatic intensity, but this effect was normal given the natural variability of a human panel at multiple levels (physiological, psychological, etc.). Thus, given these results, the panel that assessed PFC performed in a consistent and repeatable manner. For PSPPIC (Table 4), the ANOVA results led to similar conclusions. The effect of repetition and the interaction between product and panelist were insignificant for all three attributes assessed. Again, the panelist effect was also significant in all cases. Similarly, the performance of the panel for PSPPIC was therefore satisfactory in terms of repeatability and homogeneity.

#### 3.3.2. Perceived Bitterness over Baking Time in PFC and PSPPIC

For PFC, the effect of the product was not significant regarding bitterness, astringency, or global aromatic intensity (Table 3). Figure 4 represents the mean intensities for each attribute and the standard deviations of the responses. Moreover, pairwise comparisons based on the Student–Newton–Keuls (SNK) test showed no difference between the means. Thus, baking time did not have a significant impact on perceived bitterness for PFC. Hence, the differences in the saponin concentrations measured and discussed in Section 3.2 had no discernible sensory impact on the bitterness of these products. In addition, the mean scores for astringency and global aromatic intensity during baking did not indicate a trend similar to that of bitterness. This confirms that all the attributes appear to have been effectively discriminated from each other.

The sensory results for PSPPIC differed from those of PFC (Figure 5). For astringency, the product effect was not significant in the ANOVA and SNK test, so it can be concluded that baking time did not have a significant impact on the perception of astringency in PSPPIC. However, regarding bitterness, the ANOVA showed a significant product effect (α = 0.048 after removing replicate and position factors; see Appendix C), even though the SNK test did not categorize these mean intensities as being significantly different. Visually, there was a clear downward trend as a function of baking time. This is interesting because these results were expected for PFC but not for PSPPIC. Moreover, in terms of global aromatic intensity, the ANOVA indicated no significant product effect, but the SNK enabled the grouping of products into three subgroups. The different trends observed for bitterness (which gradually decreased over time) and global aromatic intensity (which was initially rated as high) then decreased (PSPPIC20) and finally increased again after 25 and 30 min, indicating significant independence in the discrimination of these attributes.

## 4. Discussion

This study aimed to characterize the fate of saponins in pea ingredients undergoing different degrees of fractionation as well as in baked products made using these more or less refined flours. Sponge cake-like products were used as models to illustrate the applicability of legume ingredients in traditional foods and to study the chemical transformations induced by the process that could have an impact on the quality perceived by consumers. As saponins are bitter compounds, it was interesting to be able to relate differences in chemical composition to a human sensory perception of bitterness. Indeed, bitterness is generally described as an obstacle to the use of pea flour, but its evolution under the effect of processing remains unexplored. Saponin βg was found in significantly higher quantities in unrefined pea flour (PF), while Bb was clearly dominant in PPI. Furthermore, PPI displayed the highest concentration of saponin among the pea ingredients, whereas PS contained very low levels of both compounds. This result illustrates the impact of refining which removes a large quantity of the native saponin. Other studies carried out on the same type of ingredients also reported that the fractionation process was primarily responsible for inactivating endogenous enzymes such as lipoxygenase and for changes to the chemical composition of reactive precursors, impacting the aroma profile and functionality of PFC compared to PSPPIC [27].

Based on this knowledge, the selection of ingredients for foods can be optimized toward flavor quality, functionality, and environmental sustainability. The use of a reformulated flour based on purified fractions (PS + PPI, 2:1 *w*/*w*, designed to have the same starch:protein ratio as in the flour), therefore appears to be a good strategy to enable a 50% reduction in the amount of bitter pea components, such as saponins, when compared to a raw pea flour. Nevertheless, this new ingredient will be less sustainable than raw flour and will not perform well under life cycle assessment, so a comprehensive cost/benefit analysis should be carried out.

Moreover, if the heat treatment applied to food could in itself degrade the most bitter βg, this would be a clever way to use less refined ingredients (i.e., PF) that might still be better accepted by consumers of the end products. The kinetic results on PFC indeed showed that during baking, saponin βg levels fell, while those of Bb rose, thus confirming the results on βg thermolability obtained using simple and liquid model systems [11]. These results, therefore, show that, in the case of processed foods containing different reactive compounds, the reactivity potential of the pea ingredients should be taken into account during the different processing steps in order to optimize the product and process. Interestingly, PFC and PSPPIC displayed quite similar total amounts of saponins after a standard baking time (25 min at 170 °C, Figure 2) that were much closer than in the original ingredients (Figure 1).

However, the results of this study show that the sensory perception of bitterness after baking varied depending on the degree of refinement of the pea flour (whole flour or flour reconstituted from protein isolates and starch); it decreased significantly during baking for PSPPIC but not for PFC. Indeed, the clear decrease observed for saponin βg during baking might have been too subtle for sensory discrimination in PFC, and we cannot exclude that other bitter compounds might also have participated in sensory perception [15,16,17] and to a different degree between PFC (with a more complex initial chemical composition) and PSPPIC. In addition, other factors not considered might explain this decrease in the bitter sensation of PSPPI, such as protein denaturation and a consequent reduction in saponin bioavailability. Recovery studies might have helped to obtain a more accurate estimate of the original saponin concentration, but the results presented in this study only considered a readily available fraction of saponins, while those bound to proteins or other ingredients may have been overlooked.

A clearer understanding of these interactions both in raw materials and complex foods, and experiments to investigate saponin–protein interactions and the degree of protein denaturation–hydrolysis in cakes during baking may constitute future avenues of research to understand the bitter perception of saponins in pea-based foods. Moreover, the interactions between food formulation and processing factors, and their effects on other flavor compounds related to ingredient reactivity, should also be addressed in future studies.

## 5. Conclusions

This study shows that flavoring substances can be tracked from raw material to final product in order to understand the impact of relevant steps in the product development process. The present study opens avenues for the exploration of pea saponins and other bitter compounds in a wider range of products, and its results will provide knowledge and expertise not only in more traditional foods such as pasta or cakes, where protein-rich legume flours have become promising alternatives for fortified and gluten-free foods, but also in new and increasingly popular foods, such as meat analogues or dairy alternatives. As the development of these foods progresses, these studies may be useful for the design of foods focused on quality and process sustainability.

## Figures and Tables

**Figure 1 foods-11-02919-f001:**
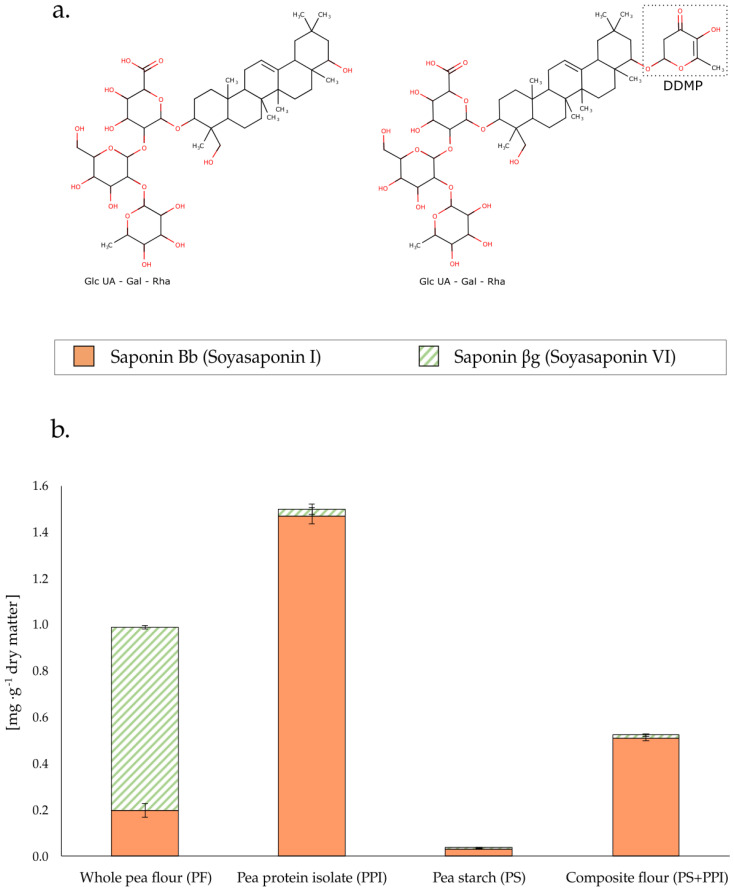
(**a**). Chemical structures of saponin Bb (on the left) and saponin βg (on the right). βg presents 2,3-dihydro-2,5-dihydroxy-6-methyl-4H-pyran-4-one (DDMP) attached to the aglycone C22, while in Bb this pyranone is substituted by a hydroxyl group. (**b**). Quantities of the two saponins measured in the flours (PF and PS + PPI) and the individual fractions (PPI and PS). Saponin concentrations in PS + PPI were calculated considering PS + PPI as a 2:1 mixture (*w*/*w*) of PS and PPI. Therefore, the resulting saponin concentrations in PS + PPI were determined using the following equation: C_PS+PPI_ = 2 × C_PS_ + 1 × C_PPI_, where C represents the concentration of either one of the saponins quantified. Saponin βg is represented in striped green and saponin Bb in orange.

**Figure 2 foods-11-02919-f002:**
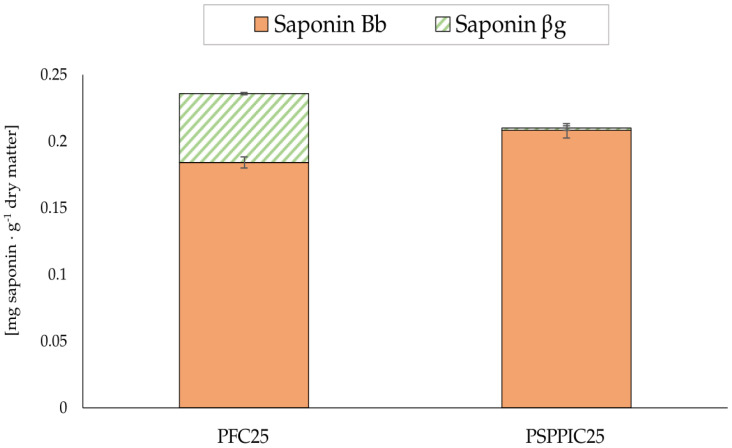
Comparison between the saponins extracted from cakes baked at 170 °C for 25 min: PFC25 = pea flour cake and PSPPIC25 = reconstituted flour cake made of PS + PPI. Saponin βg is represented in striped green and saponin Bb in orange.

**Figure 3 foods-11-02919-f003:**
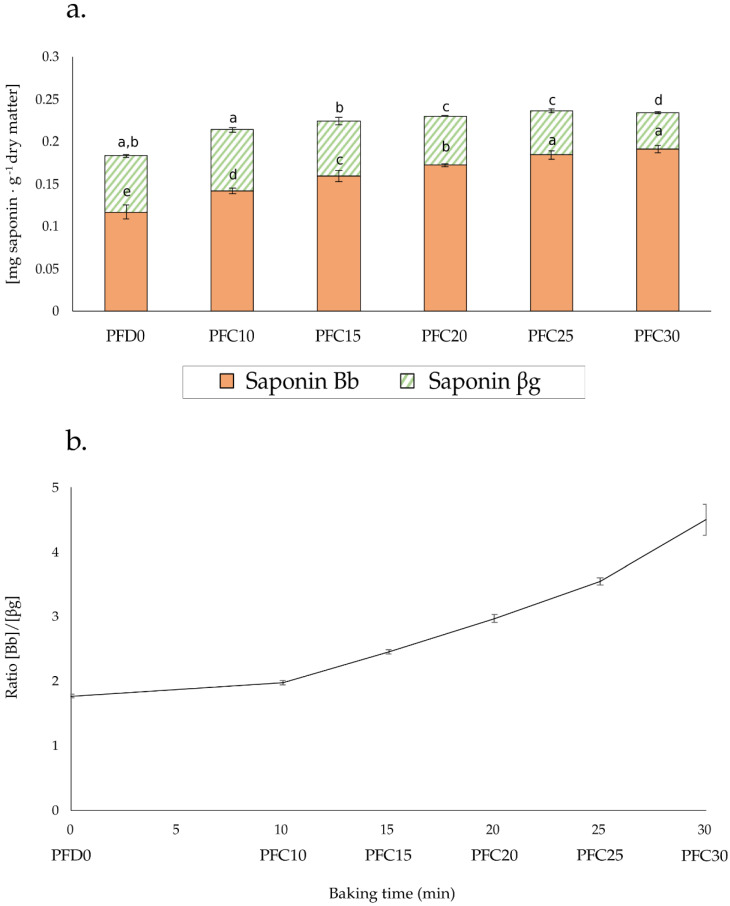
Saponins measured in the dough and in cakes baked at five different baking times. (**a**) Concentrations [mg·g^−1^ dry matter] of saponin βg (in striped green) and Bb (in orange) in PFC as a function of baking time. Different letters at the top of each color bar represent significantly different means with respect to each saponin type across samples (*p* < 0.05). PFD0 = pea flour unbaked dough, PFC10 = pea flour cake baked for 10 min, PFC15 = pea flour cake baked for 15 min, PFC20 = pea flour cake baked for 20 min, PFC25 = pea flour cake baked for 25 min, and PFC30 = pea flour cake baked for 30 min. (**b**) Bb:βg ratio plotted versus baking times (min). The corresponding cakes are also specified on the X-axis.

**Figure 4 foods-11-02919-f004:**
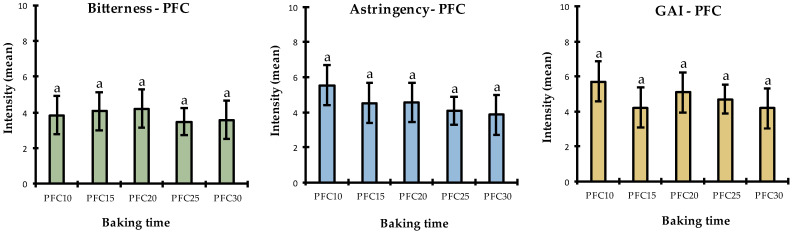
Mean intensities of the three attributes (bitterness, astringency, and aroma) assessed by the sensory panel in PFC samples over baking time. Significantly different means (*p* < 0.05) are represented by letters above each bar. PFC10 = pea flour cake baked for 10 min, PFC15 = pea flour cake baked for 15 min, PFC20 = pea flour cake baked for 20 min, PFC25 = pea flour cake baked for 25 min, and PFC30 = pea flour cake baked for 30 min. GAI = global aromatic intensity.

**Figure 5 foods-11-02919-f005:**
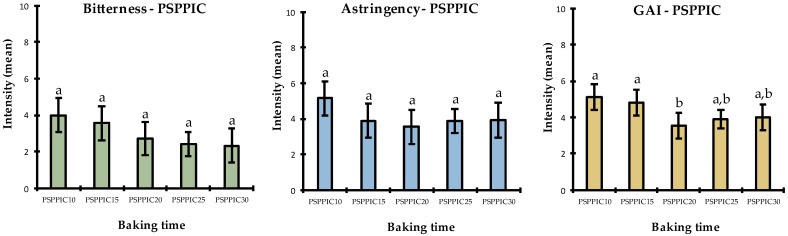
Mean intensities of the three attributes (bitterness, astringency, and aroma) assessed by the sensory panel in PSPPIC samples over baking time. Significantly different means (*p* < 0.05) are represented by letters above each bar. PSPPIC10 = pea flour cake baked for 10 min, PSPPIC15 = pea flour cake baked for 15 min, PSPPIC20 = pea flour cake baked for 20 min, PSPPIC25 = pea flour cake baked for 25 min, PSPPIC30 = pea flour cake baked for 30 min, and GAI = global aromatic intensity.

**Table 1 foods-11-02919-t001:** Cakes, baking conditions, and types of analyses (X = determined).

Labels	Baking Conditions and Formula	Analyses Performed	
		Saponin Analysis	Descriptive Sensory Analysis
PFD0 (dough before baking)	0 min, 170 °C, PF	X	
PFC10	10 min, 170 °C, PF	X	X
PFC15	15 min, 170 °C, PF	X	X
PFC20	20 min, 170 °C, PF	X	X
PFC25	25 min, 170 °C, PF	X	X
PFC30	30 min, 170 °C, PF	X	X
PSPPIC10	10 min, 170 °C, PS + PPI		X
PSPPIC15	15 min, 170 °C, PS + PPI		X
PSPPIC20	20 min, 170 °C, PS + PPI		X
PSPPIC25	25 min, 170 °C, PS + PPI	X	X
PSPPIC30	30 min, 170 °C, PS + PPI		X

**Table 2 foods-11-02919-t002:** Sensory attributes explored in the quantitative descriptive analysis within their sensory categories, scale and reference samples used in the training sessions.

Category	Attributes	Attributes in French	Scale	Reference Sample
Taste	Bitter	Amer	0–10	Pea flour suspension (10–25 g·L^−1^) Caffeine solution (0.6–3 g·L^−1^)
Mouthfeel	Astringency	Astringence	0–10	Pea flour suspension (10–25 g·L^−1^) Aluminum and potassium sulfate solutions (2–6 g·L^−1^)
Aroma	Global aromatic intensity (GAI)	Intensité aromatique globale	0–10	n.d.

n.d.: Not Determined. * Attributes were selected beforehand based on the research questions, so attribute selection was not necessary. ** Astringency was selected to ensure that it was well discriminated from bitterness. Bitterness was assessed with the nose clipped to avoid aroma interference during consumption. GAI was also selected to ensure there was no aroma interference in the taste perception of bitterness.

**Table 3 foods-11-02919-t003:** Results of three-way ANOVA for the PFC samples (PFC10, PFC15, PFC20, PFC25, and PFC30). Significant *p*-values are in bold (α = 0.05). GAI = Global Aromatic Intensity.

PFC	Panelist		Replicate		Product		Position		Panelist × Product	
	F	*p*-Value	F	*p*-Value	F	*p*-Value	F	*p*-Value	F	*p*-Value
Bitterness	2.453	0.007	0.259	0.622	0.559	0.694	1.337	0.325	1.559	0.227
Astringency	2.280	0.013	0.465	0.511	0.957	0.438	0.582	0.714	0.665	0.840
GAI	1.913	0.040	0.074	0.791	0.858	0.495	0.613	0.693	0.683	0.825

**Table 4 foods-11-02919-t004:** Results of three-way ANOVA for PSPPIC samples (PSPPIC10, PSPPIC15, PSPPIC20, PSPPIC25, and PSPPIC30). Significant *p*-values are in bold (α = 0.05). GAI = Global Aromatic Intensity.

PSPPIC	Panelist ID		Replicate		Product		Position		Panelist × Product	
	F	*p*-Value	F	*p*-Value	F	*p*-Value	F	*p*-Value	F	*p*-Value
Bitterness	3.281	0.001	0.914	0.362	2.333	0.066	0.547	0.738	1.057	0.502
Astringency	2.375	0.009	0.449	0.518	0.680	0.609	2.870	0.073	2.473	0.061
GAI	3.122	0.001	0.500	0.496	1.629	0.179	1.237	0.361	2.327	0.074

## Data Availability

Data are contained within the article.

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
