# Peer review of "Saponins from Pea Ingredients to Innovative Sponge Cakes and Their Association with Perceived Bitterness"

_foods, 2022, doi:10.3390/foods11182919_

Round 1

Reviewer 1 Report

In this manuscript, the authors reported an interesting finding of pea bitter substance in baking cake. I don’t have major criticism, but some comments needed to be addressed.

1. Page 1, line 2: The first letter of everyword should be capitalized.

2. Page 1, lines 6-9: E-mails of all authors should be provided.

3. Page 3, Table 1: About labels, should PSPPI be replaced by PSPPIC?

4. Page 5, Table 2: Please try to place Table 2 on the same page instead of separating it on pages 5 and 6.

5. Page 6, line 211: ‘2.6’ Data alalysis should be replaced by 2.7.

6. Page 7, Figure 1: Two senstences are placed on page 8. Please try to put them together.

7. Page 11, Figure 3b: Please remove numbers (0, 5, 10, 15, 20, 25, 30) from X-axis.

Reviewer 2 Report

Hi dear

This article "Saponins from pea ingredients to novel sponge cakes and their link with perceived bitterness was revised and has a novelty and I recommend consideration of the following comments.

Title: If you can rewrite and make it more interesting for readers. I propose: “Saponins from pea ingredients to innovative sponge cakes and their association with perceived bitterness”.

Abstract:

·       Line 22-23: Please include the quantity of βg?

·       Line 24-25: please write whole pea flour (PF) instead of whole flour. Meanwhile you pointed as a complete form in line 12 and there is no need repeat it again.

·       Line 23: 1.497 mg · g-1 dry flour or dry matter.

·       The type of statistical design used in this research should be mentioned.

·       No statistical analysis was done for data pointed in the abstract why?

Keywords: It is perfect and appropriate.

Abbreviation:

·       Please provide “Abbreviation section consequent the Keywords

Introduction:

·        Line 48-49: How did you prove that which components, (i.e., saponins, polyphenols, lipid oxidation, products, protein hydrolysates, and Maillard reaction products) have been the main factor for bittering of final cake?

·        Line 72-73: the abbreviations are not correct.

·        Line 75: why did you do over baking time? It is obviously the bitterness can achieved from maillard reaction or etc. please mention to line 48-49.

Materials:

·                   Please write materials as Company Name (City, Country), especially for chemical analysis assessment which used in the study.

·                 Table 1: Why did not you apply the treatment contain the only PS and only PPI for cake making?

·     Table 1: What is the meaning of the 0 min, 170 °C, PF?

Methodology:

·       2.4 Cakes physical characterization. Please cite and use the citations as the follow:

·        

·       https://doi.org/10.1590/fst.52120 for physical characterization in line 121.

·       Line 111-121: The way of expressing the method of measuring macronutrients and other other parameters has a scientific flaw. Please take help from the following article for the correct way of expressing it, so that the standard number of the working method should be clearly stated (https://doi.org/10.1590/fst.60820).

 “Results:

·       In Figure 1. b: please include the percentage of every type of saponin.

·       Line 246-247: The standard error or standard deviation should not be used in the entire text of the results

·       All Tables: The alphabetical statistical letters for the means should all be modified such that the greatest number has the letter a and as the numbers go lower, letters b, c etc.

·       PFD0 in Fig 3a: why a,b? Please recheck it and reconsider statistical analysis for the data.

·       Table 3 and 4 i.e., ANOVA table are not crucial and important in food science and technology field but also the figures consequated are enough.

·       Fig 4 and 5: unfortunately the standard deviations are too high which resulted to no statistical significant differences between the treatments.

Discussion:

·       Discussion text must grammar improve and in some cases it is very weak and maybe there is no discussion at all.

Conclusions:

·       It was omitted. Please consider to it as scientific detailed and concise.

References: It is OK.

The article has many flaws in express and concept of English, it is suggested to be revised in a scientific and native way.
